# Phenotypic characterization of primary cardiac fibroblasts from patients with HFpEF

Yuhua Zhang[1], An O. Van Laer[1], Catalin F. Baicu[1], Lily S. Neff[1], Stanley Hoffman[2], Marc R. Katz[3], Sanford M. Zeigler[3], Michael R. Zile[1,4], Amy D. Bradshaw[1,4]*

1 Division of Cardiology, Department of Medicine, Medical University of South Carolina, Charleston, South Carolina, United States of America, 2 Division of Rheumatology, Department of Medicine, Medical University of South Carolina, Charleston, South Carolina, United States of America, 3 Department of Surgery, Medical University of South Carolina, Charleston, South Carolina, United States of America, 4 Ralph H. Johnson Veterans Affairs Medical Center, Charleston, South Carolina, United States of America

* bradshad@musc.edu

**Data Availability Statement:** All relevant data are within the manuscript and its Supporting information files.

## Abstract

Heart failure is a leading cause of hospitalizations and mortality worldwide. Heart failure with a preserved ejection fraction (HFpEF) represents a significant clinical challenge due to the lack of available treatment modalities for patients diagnosed with HFpEF. One symptom of HFpEF is impaired diastolic function that is associated with increases in left ventricular stiffness. Increases in myocardial fibrillar collagen content is one factor contributing to increases in myocardial stiffness. Cardiac fibroblasts are the primary cell type that produce fibrillar collagen in the heart. However, relatively little is known regarding phenotypic changes in cardiac fibroblasts in HFpEF myocardium. In the current study, cardiac fibroblasts were established from left ventricular epicardial biopsies obtained from patients undergoing cardiovascular interventions and divided into three categories: Referent control, hypertension without a heart failure designation (HTN (-) HFpEF), and hypertension with heart failure (HTN (+) HFpEF). Biopsies were evaluated for cardiac myocyte cross-sectional area (CSA) and collagen volume fraction. Primary fibroblast cultures were assessed for differences in proliferation and protein expression of collagen I, Membrane Type 1-Matrix Metalloproteinase (MT1-MMP), and α smooth muscle actin (αSMA). Biopsies from HTN (-) HFpEF and HTN (+) HFpEF exhibited increases in myocyte CSA over referent control although only HTN (+) HFpEF exhibited significant increases in fibrillar collagen content. No significant changes in proliferation or αSMA was detected in HTN (-) HFpEF or HTN (+) HFpEF cultures versus referent control. Significant increases in production of collagen I was detected in HF (-) HFpEF fibroblasts, whereas significant decreases in MT1-MMP levels were measured in HTN (+) HFpEF cells. We conclude that epicardial biopsies provide a viable source for primary fibroblast cultures and that phenotypic differences are demonstrated by HTN (-) HFpEF and HTN (+) HFpEF cells versus referent control.

**Funding:** YES: This work was partially supported by NIH grant 5R01-HL123478-01 to MRZ, NIH T32GM132055 and a FA9550-21-F-0003 through the National Defense Science and Engineering Graduate (NDSEG) Fellowship Program, sponsored by the Air Force Research Laboratory (AFRL), the Office of Naval Research (ONR) and the Army Research Office (ARO) to LSN and by the Department of Veterans Affairs under grants CX001608 to ADB and MRZ.

**Competing interests:** The authors have declared that no competing interests exist.

## Introduction

Heart failure affects a significant number of people in the U.S. and around the world. Heart failure is often partitioned into two categories, heart failure in individuals that have a reduced ejection fraction, referred to as HFrEF (heart failure with reduced ejection fraction), versus heart failure with a preserved ejection fraction (HFpEF) [1]. A hallmark of HFpEF is both hypertrophic growth of cardiac myocytes and the accumulation of extracellular matrix (ECM) predominated by fibrillar collagens [1]. Studies have shown that changes in myocyte structural proteins as well as increases in myocardial collagen contribute to increases in left ventricular chamber and myocardial diastolic stiffness characteristic of patients diagnosed with HFpEF [2].

Cardiac fibroblasts are the primary cell type that produce and secrete fibrillar collagens in healthy hearts. Murine models of clinically relevant heart disease provide evidence that resident fibroblasts are also the primary cell type producing fibrillar collagens in response to left ventricular pressure-overload (LVPO) [3]. In murine hearts subject to LVPO, activation of resident fibroblast populations are observed [3, 4].

Single cell transcriptomic analysis of normal human hearts by Litvinukova et al. suggested that the resident cardiac fibroblast population could be divided into 7 distinct clusters based on gene expression [5]. Furthermore, analysis of specific gene expression profiles suggested that subsets of the resident fibroblast population are responsible for fibrillar collagen production. In response to LVPO in mice, resident fibroblast populations increase in number and initiate expression of specific proteins associated with fibroblast activation [3, 6]. Activated and myofibroblast populations are predicted to produce higher levels of collagens and contribute to pro-fibrotic deposition of ECM fibrillar collagens [7].

Characterization of primary human cardiac fibroblasts are limited by availability of human heart tissue. Most studies to date have utilized cells that originate from end stage hearts undergoing transplant or unused donor hearts, that represent either healthy or terminally diseased cardiac tissue. To our knowledge, this is the first study performed using primary human cardiac fibroblasts derived from LV epicardial patient biopsies undergoing cardiac surgical procedures divided into 3 groups: referent control, hypertensive heart disease without heart failure (HTN (-) HFpEF), or hypertensive heart disease with HFpEF (HTN (+) HFpEF).

## Materials and methods

### Study population

The study cohort consisted of patients recruited to undergo intraoperative LV myocardial biopsy from among those scheduled for elective coronary artery bypass grafting (CABG) at the Ralph H. Johnson Department of Veterans Administration Medical Center and the Medical University of South Carolina Hospital Authority in Charleston, SC between January, 2008 and January, 2020 and who satisfied the inclusion and exclusion criteria specified below, and previously published. All patients signed written consent forms approved by the MUSC institutional review board. Demographic, medication, and laboratory data and cardiac catheterization results (coronary anatomy, LV end-diastolic pressure) are presented in Table 1. The severity of CAD was graded based on the number of major vessels (left anterior descending, left circumflex, right coronary arteries) with a stenosis >70%, with left main coronary stenosis considered as 2 vessels. All studies were reviewed and approved by the MUSC institutional review board.

**General inclusion criteria.** Patients scheduled to undergo CABG over 21 years of age, with a LV ejection fraction ≥50%, normal wall motion, LV end-diastolic volume index <75

Table 1. Demographic, structural, and hemodynamic data.

| | Referent Control | HTN + LVH | |
| --- | --- | --- | --- |
| | | (-) HFpEF | (+) HFpEF |
| Age (years) | 65 ± 9 | 68 ± 9 | 72 ± 8 |
| Sex (F/M) | 2/5 | 1/5 | 6/6 |
| BSA ($m^2$) | 1.9 ± 0.1 | 2.0 ± 0.2 | 1.9 ± 0.2 |
| SBP (mmHg) | 123 ± 17 | 137 ± 13 * | 141 ± 20 * |
| DBP (mmHg) | 74 ± 6 | 78 ± 10 | 79 ± 11 |
| HR (bpm) | 78 ± 15 | 73 ± 10 | 73 ± 13 |
| LV $EDV_i$ ($mL/m^2$) | 48 ± 13 | 52 ± 15 | 55 ± 17 |
| LV $ESV_i$ ($mL/m^2$) | 15 ± 6 | 13 ± 5 | 19 ± 7 |
| LV EF (%) | 68 ± 8 | 67 ± 9 | 66 ± 7 |
| LV $Mass_i$ ($g/m^2$) | 92 ± 28 | 152 ± 20 * | 141 ± 14 * |
| E (cm/sec) | 76.6 ± 12.5 | 70.0 ± 12.8 | 93.9 ± 12.4 *# |
| e' (cm/sec) | 10.5 ± 2.6 | 9.2 ± 0.7 | 8.7 ± 1.9 *# |
| E/e' | 7.9 ± 3.4 | 7.7 ± 1.7 | 11.6 ± 2.3 *# |
| PCWP (mmHg) | 12.2 ± 4.4 | 12.0 ± 2.2 | 17.0 ± 3.5 *# |
| LA $Vol_i$ ($mL/m^2$) | 22.8 ± 5.7 | 19.4 ± 7.4 | 41.5 ± 14.1 *# |

Abbreviations:

Data = Mean ± St.Dev;

*p<0.05 vs Referent Control;

#p<0.05 vs HTN+LVH (-)HFpEF;

HTN = hypertension; LVH = LV hypertrophy; LV = left ventricle; BSA = body surface area; SBP = systolic blood pressure; DBP = diastolic blood pressure; HR = heart rate; LV = left ventricular; $EDV_i$ = end diastolic volume index; $ESV_i$ = end systolic volume index; EF = ejection fraction; E = transmitral Doppler early filling velocity; e' = tissue Doppler early diastolic mitral annular velocity; PCWP = Pulmonary capillary wedge pressure; LA = left atrium; CVF = collagen volume fraction.

$mL/m^2$, and without evidence of previous myocardial infarction were eligible. Patients were categorized into 3 groups as specified below: referent control, HTN(–)HFpEF, and HTN(+) HFpEF.

**Specific patient group inclusion criteria.** Referent control patients fulfilled the general inclusion criteria above, and did not have a history of hypertension or diabetes.

HTN(-)HFpEF patients fulfilled the general inclusion criteria above and had a history of hypertension documented in their records and/or had been told of this diagnosis by a physician, and/or were receiving medications for its treatment. These patients had no evidence of heart failure as defined below.

HTN(+)HFpEF patients fulfilled the general inclusion criteria above and had hypertension and HFpEF as specified by the European Society of Cardiology and Heart Failure Society of America criteria. These criteria require: (1) signs and symptoms of heart failure (Framingham or Boston criteria, exercise testing, quality of life questionnaire), (2) LV ejection fraction ≥50%, (3) LV end-diastolic volume index <75 $mL/m^2$, (4) evidence of diastolic LV dysfunction obtained invasively (cardiac catheterization) or noninvasively (transmitral or tissue Doppler or left atrial size), and (5) exclusion of noncardiac diseases that could cause symptoms commonly present in patients with heart failure.

**Exclusion criteria.** Patients were excluded if they had a previous ST segment elevation myocardial infarction, LV ejection fraction <50%, LV end-diastolic volume index >75 mL/ $m^2$, significant valvular or other noncoronary heart disease, severe chronic pulmonary disease requiring oral steroids and continuous oxygen therapy, any noncardiac disease or condition known to affect myocardial function, anemia (hemoglobin <13.0 g/dL), serum creatinine

>2.0 mg/dL, poorly controlled hypertension (blood pressure >140/90 mm Hg), off-pump or emergency CABG, morbid obesity, history of substance abuse, inability to provide informed consent, poorly controlled diabetes (hemoglobin A1c >8.5% within the past 6 months), active malignancy, severe connective tissue disease, severe liver disease, hypertrophic cardiomyopathy, restrictive cardiomyopathy or constrictive pericarditis, human immunodeficiency virus or active infection.

## Myocardial biopsy procedure

One anterior free wall epicardial—midwall biopsy (~20–100) mgs was collected from each patient during intraoperative procedures after the patient was placed on cardiopulmonary bypass. All patients were followed until discharge, with particular attentions to ventricular arrhythmias and bleeding complications. No adverse effects or post-operative complications ascribable to participation in the study were detected. Fresh biopsies were typically divided for fixation, embedding and histochemical analysis, and for fibroblast isolation.

## Histology and immunohistochemistry

One section of each biopsy was fixed in Zinc-formalin, embedded in paraffin, and sectioned on a microtome. Picrosirius red staining was carried out as previously described [8]. Each biopsy was viewed under polarized light to capture the birefringence generated by the fibrillar collagen present in the section. Collagen volume fraction (CVF) was then calculated from five separate regions of interest (RIO) from three separate sections taken from each biopsy using SigmaScan Pro and presented as percent collagen content of the total area in the ROI [8]. Wheat germ agglutinin conjugated to peroxidase (Vector Labs) was incubated on myocardial sections and detected using tyramide amplification (Biotium). Stained sections were visualized on a Keyence BZ-X800 fluorescent microscope which allows for reconstruction of stitched images such that the entire region of the tissue section can be viewed in one image. Three fields per biopsy were evaluated that contained myocytes with cross-sectional orientation, approximately 500 cells per field were quantified (~1500 cells per biopsy). One section per biopsy was used to evaluate CSA. Myocyte cross sectional area (CSA) was calculated using the Hybrid Cell Count function (Keyence BZ-X800 Analyzer).

## Primary cell isolation

Tissue from each biopsy was finely minced and incubated with successive changes of collagenase (Liberase Blendzyme 3, Roche) over one to two hours with intermittent trituration until digestion was complete. Pooled collagenase fractions were placed in growth media (Fibroblast Media 2, Promocell, 10% fetal calf serum, antibiotics/antimycotics) to halt collagenase digestion and then subjected to centrifugation for five minutes at room temperature. Collagenase containing media was removed and cell pellets resuspended in growth media. Cultures were plated at 37°C. Nonadherent cells were removed ~ 12–18 hours after initial plating. Cells proliferation assays were carried out in P1 or P2 and protein production analyses were performed using cell passages 2 and 3 to allow for sufficient expansion of fibroblast cultures to generate sufficient numbers to evaluate function.

## Proliferation assays

Proliferation assays were typically carried out on primary isolations as previously described [9]. Briefly, each primary cell preparation was plated at equal density in triplicate and allowed to attach in 10% fetal calf serum-containing media for ~ 8 hours. Proliferation assays were

performed at P1 or P2 for each isolation. Cells were then starved overnight in serum free media containing 1% bovine serum albumin (BSA). Following starvation, cells were either stimulated with 2% FCS containing growth media or retained in 1% BSA for 8 hours to quantify increases in proliferation in response to stimulation with FCS. Cells were then incubated with 2 μCi/ml $^3$H-thymidine for 18 hours. Following incubation, media was removed and cell layers washed with 10% tri-chloro acetic acid (TCA) and then solubilized in 0.1 N NaOH for 30 minutes at 65 C. $^3$H-thymidine incorporation was measured in each condition with a Beckman Scintillation counter and averaged across triplicates. Proliferative index was calculated as the average percent increase in cells treated with 2% FCS over average control BSA cultures.

## Immunoblot analysis

Protein production from primary cell isolates were assessed by immunoblot analysis as previously described [10]. Protein production was evaluated in cells from P2 or P3. Briefly, ~3 X 10$^5$ cells were seeded in tissue culture dishes in growth media and conditioned media and cell layers were collected at day 2 and 3 (and some cases 4 when allowable cell numbers were available) after plating. Cell layers were washed one time in PBS, collected in 1% deoxycholate with protease inhibitors (protease complete, Roche), and centrifuged at 10,000 x g. Proteins soluble in deoxycholate were separated by SDS-PAGE analysis and subjected to immunoblot analysis. Primary antibodies used were rabbit polyclonal anti-C-terminal propeptide of collagen I (LF-41, a kind gift of Dr. Larry Fisher, NIH) [11], rabbit polyclonal anti-telopeptide of collagen I (generated and verified at MUSC), rabbit polyclonal MTI-MMP (Chemicon, #AB8221 & Abcam, #ab38971), and mouse monoclonal against alpha smooth muscle actin (Sigma, #A2547). Antibodies against tubulin were used to establish equal loading of protein and to serve as a readout of baseline protein production for each culture. Primary antibodies were detected with appropriate secondary antibodies conjugated to horse radish peroxidase incubated with ECL reagent. Detection was captured using either X-ray film or a GE image documentation center.

## Statistical analysis

Data are reported as Means SD in tables or graphically depicted as box-and-whisker diagram (KaleidaGraph 4.5.3; Synergy Software, Reading, PA). To detect significant differences between all measured variables, we performed student's t-test or one-way ANOVA followed by Bonferroni test for pairwise multiple comparisons (SigmaStat 4.0; Systat Software Inc., San Jose, CA). A $P$ value of $<0.05$ was considered significant.

## Results

### Demographic, structural, and hemodynamic data

Data presented in Table 1 indicate that the patients recruited into the 3 stated patient groups were in compliance with the eligibility criteria for each group. Across the 3 groups there were no differences in age, sex, race or Body Surface Area (BSA), LV volume or EF. In the HTN (-) HFpEF group, systolic blood pressure (SBP), LV mass were higher than control but not different from hypertension with HFpEF. Finally, the HTN (+) HFpEF group had increased indices of increased filling pressure (E,e', PCWP, LA vol) compared to control and HTN (-) HFpEF.

### Quantification of cell hypertrophy and tissue fibrosis

A portion of each biopsy was taken for histochemical analysis and a representative set of tissue sections from each group were assessed to determine cardiac myocyte cross-sectional area

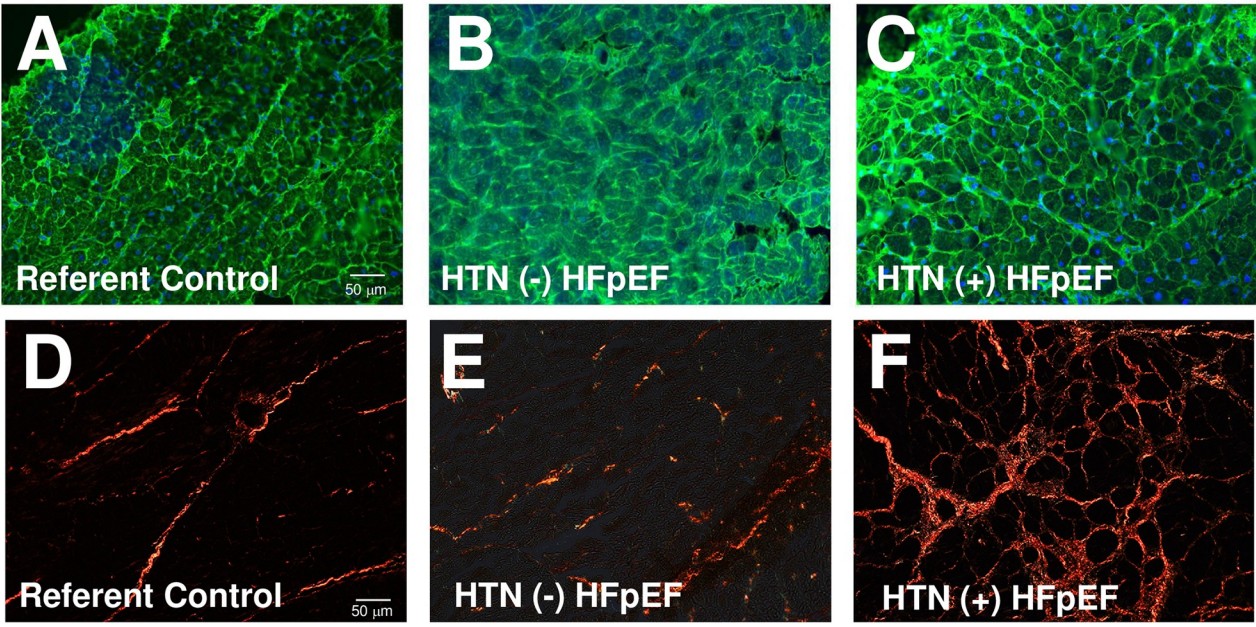

**Fig 1.** Representative sections from epicardial biopsies stained with wheat germ agglutin (WGA) were used to assess cardiomyocyte cross-sectional area (CSA) from patients categorized as A) Referent control, B) HTN (-) HFpEF, and C) HTN (+) HFpEF. Collagen volume fraction was assessed by picro-sirius red (PSR) stained tissue sections from D) Referent control, E) HTN (-) HFpEF, and F) HTN (+) HFpEF biopsies. Scale bar in A and D are indicative of magnification in each panel.

(CSA) and collagen volume fraction (CVF). Myocardial sections were stained with wheat germ agglutinin (WGA) to demarcate myocyte cell borders. Representative images are shown in Fig 1A–1C and quantified in Fig 2A. Cardiomyocyte CSA increased in sections of LV myocardium from individuals diagnosed with HTN (-) HFpEF and in biopsies from HTN (+) HFpEF in comparison to that of referent control values. CVF was assessed in the same biopsies by quantification of picrosirius red stained images (Fig 1D–1F). In contrast to myocyte CSA, CVF was not significantly increased in HTN (-) HFpEF compared with referent control (Fig 2B). However, CVF was elevated in biopsies from HTN (+) HFpEF compared to referent control and HTN (-) HFpEF (Fig 2B).

## Human primary fibroblast proliferation

Successful culturing of primary cardiac fibroblasts occurred in ~ 90% of the biopsies collected as described in Materials and Methods, although some biopsies yielded higher number of cells primarily dependent upon biopsy size. To determine whether quantifiable differences in proliferation were demonstrated in fibroblasts from different patient populations, $^3$H-thymidine incorporation assays were carried out. To compare differences in proliferation across multiple cell isolates, each primary cell culture was assessed by percent change in $^3$H-thymidine incorporation from baseline quiescent cells versus those stimulated with 2% fetal calf serum so that each isolation served as its own control. No significant differences in proliferation were observed between groups (Fig 3). However, the HTN (-) HFpEF group demonstrated a greater range in values for the proliferative indexes than either referent control or HTN (+) HFpEF perhaps indicative of greater variability in disease state in this intermediate group. A slight trend toward reduced rates of proliferation were observed in the HTN (+) HFpEF group but did not reach statistical significance.

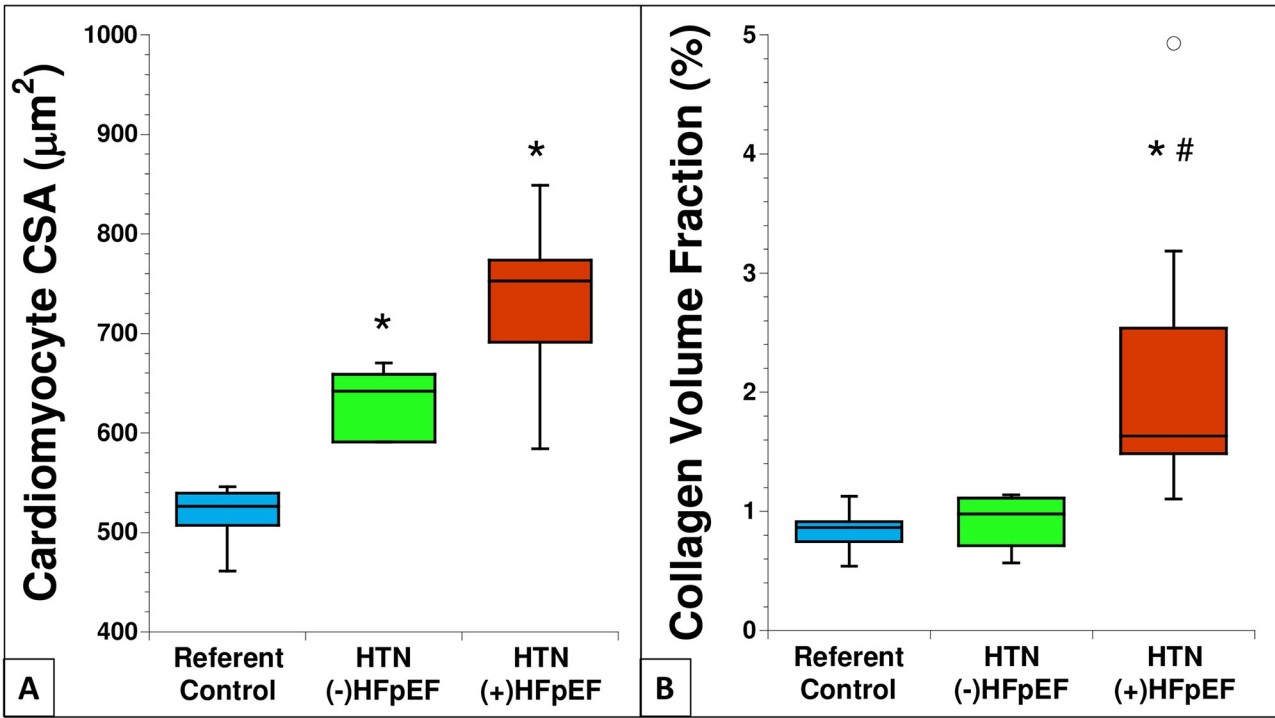

**Fig 2.** A) Quantification of cardiomyocyte CSA (Blue: Referent control (n = 7), Green: HTN (-) HFpEF (n = 6), Red: HTN (+) HFpEF (n = 12)). B) Quantification of collagen volume fraction (Blue: Referent control (n = 7), Green: HTN (-) HFpEF (n = 6), Red: HTN (+) HFpEF (n = 12)). *p<0.05 versus referent control; #p<0.05 versus HTN (-) HFpEF. Open circles represent outlier values.

## Protein production

**Collagen.** Primary cardiac fibroblasts were evaluated for procollagen production and processing by immunoblot analysis. Two separate polyclonal antibodies against collagen I were used, a polyclonal antibody generated against the C-propeptide of collagen I and a polyclonal antibody generated against the telopeptide region of collagen I. Both antibodies yielded similar results. A representative immunoblot for collagen I is shown in Fig 4. Collagen I production assessed in conditioned media by referent control, HTN (-) HFpEF and HTN (+) HFpEF at multiple days in culture are shown and demonstrate elevated levels of collagen production in HTN (-) HFpEF versus that of referent control and that of HTN (+) HFpEF. Shown in panel A are quantitative data collected from an n of four separate primary isolates for each category at day 3 (see Materials and methods, similar results were seen for days 2 and 4). In fibroblasts from patients with HTN (-) HFpEF there was a significantly higher level of total collagen I (i.e. procollagen plus pC collagen (partially processed) plus collagen I compared to referent control and HTN (+) HFpEF. Although there was a trend toward higher total collagen in HTN (+) HFpEF compared to referent control, these values did not reach statistical significance. Differences in procollagen processing, i.e. removal of C and N-terminal propeptides, were not found in either HTN (-) HFpEF or HTN (+) HFpEF compared with referent control (Fig 4B). Procollagen processing was assessed as the ratio of procollagen 1 vs total collagen 1 (procollagen + pC collagen + collagen) and expressed as percent procollagen I. Similar trends in levels of collagen production between groups were observed in blots using cell layers (S1 Fig).

**MT1-MMP.** Changes in myocardial collagen deposition are influenced both by increases in procollagen production and decreases in fibrillar collagen degradation. Levels of

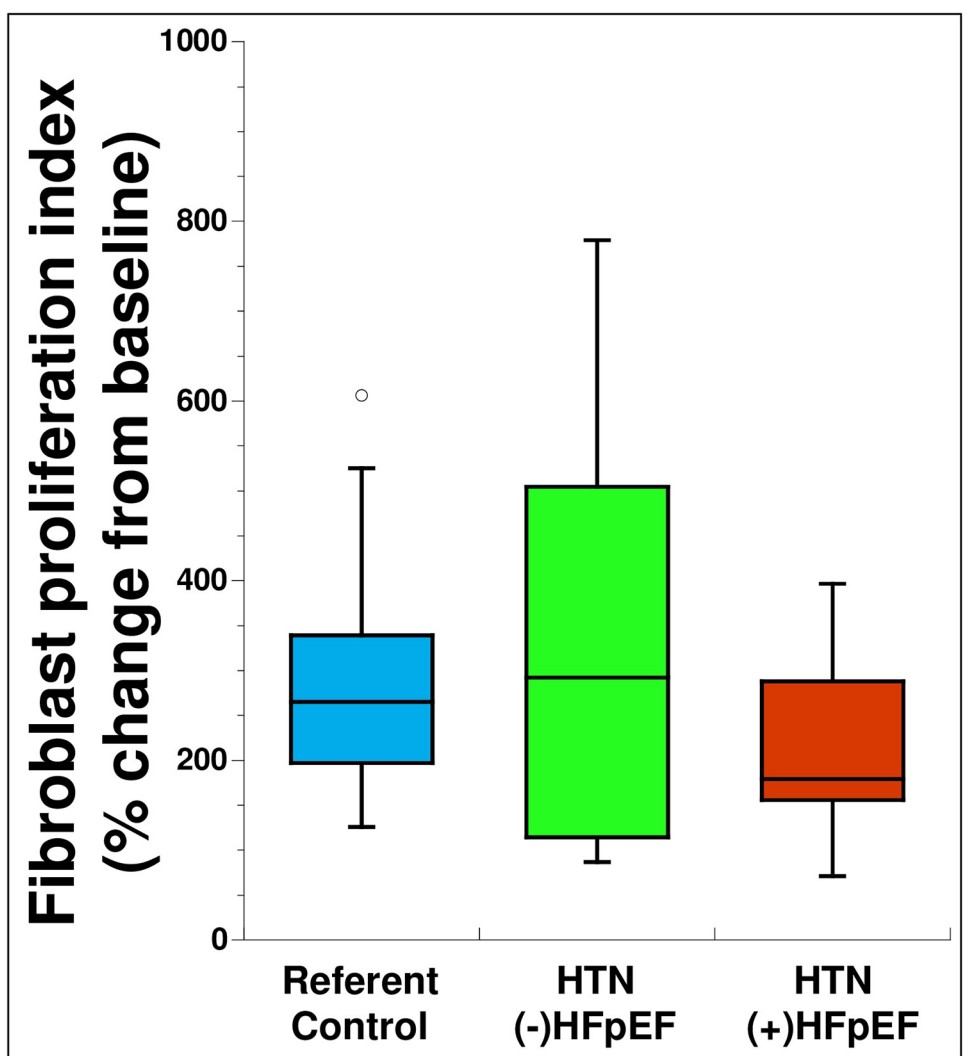

**Fig 3. Fibroblast proliferation was assessed as described in materials and methods.** No differences in proliferation were detected between the three groups. Proliferation assays were performed at P1 or P2 for each isolation. Blue: Referent control (n = 19), Green: HTN (-) HFpEF (n = 12), Red: HTN (+) HFpEF (n = 17). Open circles represent outlier values.

MT1-MMP, a collagenase implicated in cardiac ECM remodeling, was assessed in primary cardiac fibroblast cultures. Shown in Fig 5 is a representative immunoblot detecting expression of MT1-MMP in primary fibroblasts from referent control, HTN (-) HFpEF, and HTN (+) HFpEF. MT1-MMP production relative to tubulin in primary fibroblast cultures was lower in HTN (+) HFpEF cells in comparison to referent control and HTN (-) HFpEF fibroblasts.

**α Smooth muscle actin.**  α Smooth muscle actin (αSMA) is widely used as a marker of activated cardiac fibroblasts. Immunoblot analysis was used to assess levels of αSMA in HTN (+) HFpEF versus referent control and HTN (-) HFpEF. No significant differences in αSMA production were observed between primary cells from each category across three days in culture (Fig 6).

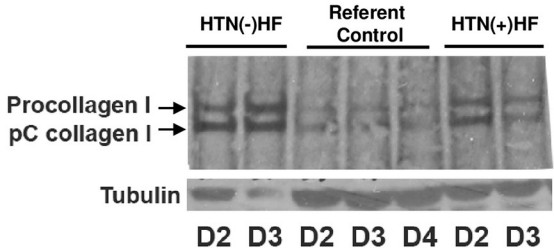

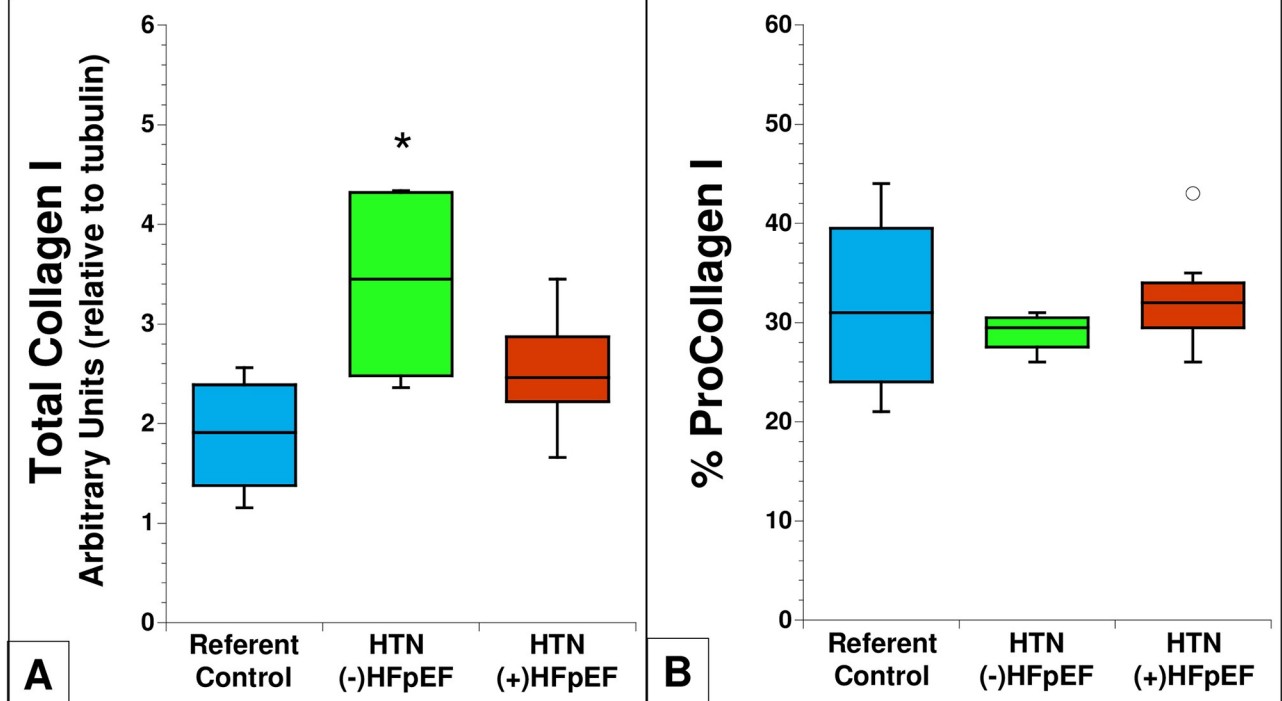

**Fig 4. Collagen production by primary fibroblasts was quantified by immunoblot analysis and determined relative to expression of tubulin.** A representative blot of collagen I bands in conditioned media is shown. Quantification of total collagen I (summary values for all collagen I bands versus tubulin) and the percent procollagen I (procollagen I band intensity/total collagen I bands) is shown. Protein production was evaluated in cells from P2 or P3. Blue: Referent control (n = 5), Green: HTN (-) HFpEF (n = 4), Red: HTN (+) HFpEF (n = 6). *p<0.05, open circles represent outlier values.

## Discussion

Cardiac fibroblasts are considered to be the primary cell type responsible for fibrillar collagen production and deposition in the myocardium. As treatments for limiting and/or regressing cardiac fibrosis represents a critical unmet need, a better characterization of changes in fibroblast phenotype induced by LVPO will improve strategies to design new therapies. The results presented here support the following, 1) LV biopsies obtained from patients with and without hypertensive heart disease can be successfully obtained and yield sufficient numbers of primary fibroblasts to characterize phenotype, 2) fibroblasts isolated from HTN (+) HFpEF patients exhibit significant differences in comparison to both HTN (-) HFpEF and to referent controls, and 3) changes in fibroblast phenotype measured *in vitro* contribute to a better understanding of the underlying mechanisms that result in the changes in ECM homeostasis and structure that occur *in vivo* in the LV myocardium of LVPO and HFpEF.

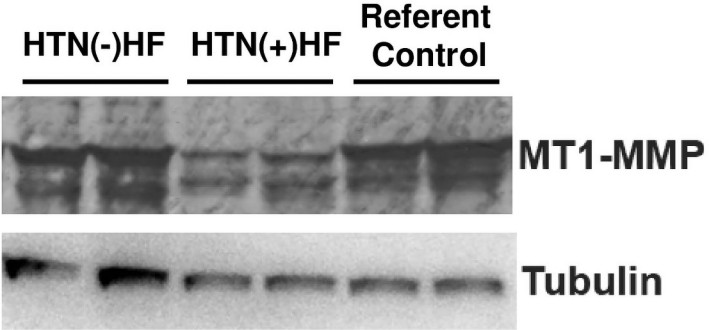

Fig 5. MT1-MMP production by primary fibroblasts was quantified by immunoblot analysis and determined relative to expression of tubulin. A representative blot of MTI-MMP in cell layers is shown. Protein production was evaluated in cells from P2 or P3. Blue: Referent control (n = 8), Green: HTN (-) HFpEF (n = 7), Red: HTN (+) HFpEF (n = 12). *p<0.05, open circles represent outlier values.

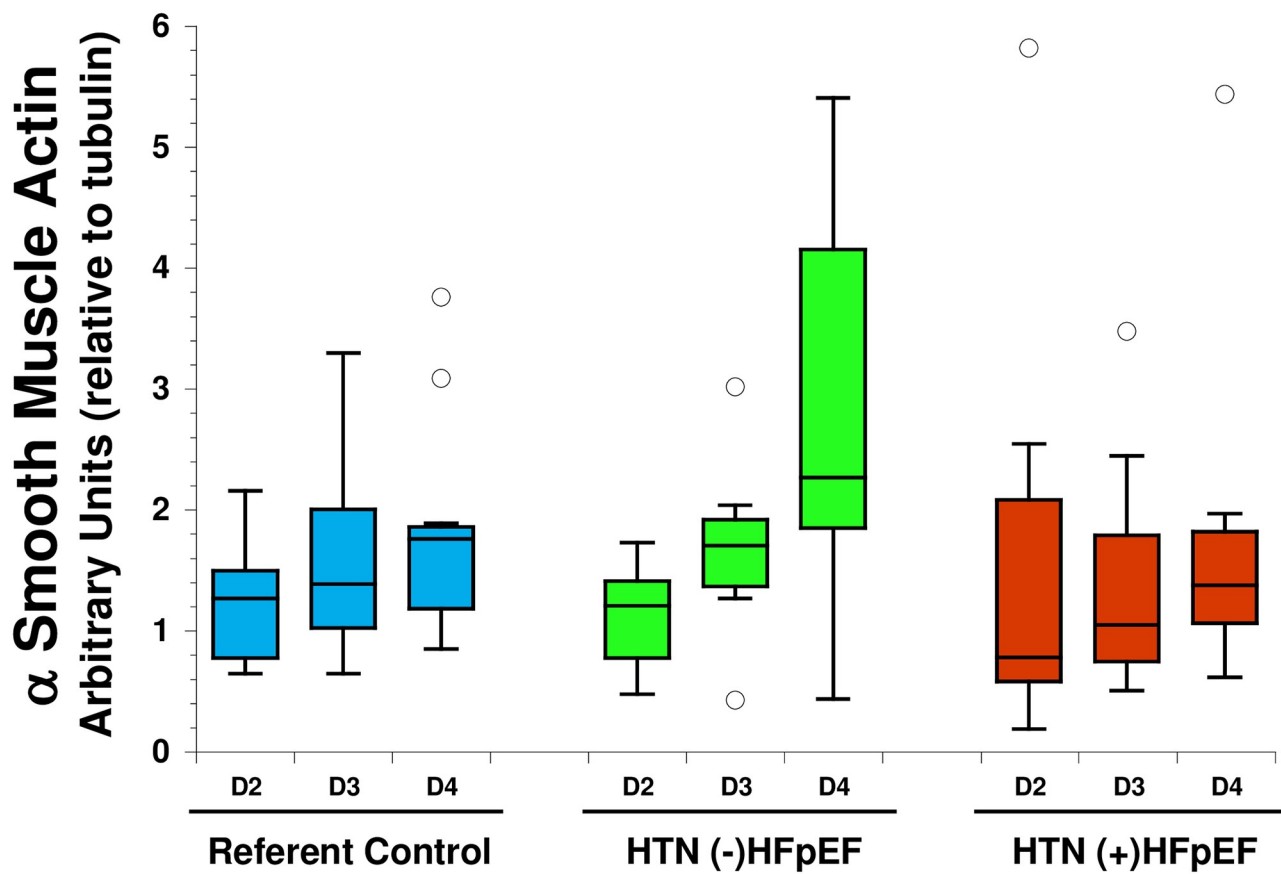

**Fig 6. αSMA production by primary fibroblasts was quantified by immunoblot analysis and determined relative to expression of tubulin at day 2, day 3, and day 4.** Protein production was evaluated in cells from P2 or P3. Blue: Referent control (n = 10), Green: HTN (-) HFpEF (n = 7), Red: HTN (+) HFpEF (n = 10). *p<0.05, open circles represent outlier values.

Significant increases in procollagen production were observed in HTN (-) HFpEF fibroblasts versus both referent control and HTN (+) HFpEF cells. Previous publications from our laboratory have reported an increase in procollagen production in a murine model of LVPO using transverse aortic constriction (TAC) [8, 12]. These changes in procollagen synthesis occur at the earliest measured time point after TAC (3 days) and precede by 2 weeks the increases in procollagen processing, insoluble collagen and the development of fibrosis in PO hearts that are present at 4 weeks [12]. Similar findings occurred in a feline RV constriction model of PO [13]. Hence, the increase in procollagen production by HTN (-) HFpEF fibroblasts might reflect a similar increase in procollagen expression that precedes collagen deposition in human hearts. However, the changes in fibroblast phenotype do not explain the increases in procollagen processing and collagen content that occur after prolonged LVPO. In the murine model, we reported one determinant of increased collagen deposition was dependent upon increases in myocardial macrophages recruited in response to PO which would not be represented in the *in vitro* fibroblast system reported here [14].

No significant differences in proliferation, procollagen processing, and αSMA production were found in HTN (+) HFpEF fibroblasts in comparison to referent control in our culture conditions. However, production of the collagenase, MT1-MMP, was found to be consistently reduced in HTN (+) HFpEF fibroblasts versus referent control and HTN (-) HFpEF cells.

Decreases in collagenase activity lead to decreased rates of collagen degradation, and increased rates of collagen accumulation, consistent with increased amounts of fibrillar collagen indicative of HTN (+) HFpEF LV epicardial biopsies. Westermann et al. reported a decrease in MMP-1, another collagenase, in biopsies from a patient cohort similar to that reported here, classified as heart failure with a normal ejection fraction [15]. Further, primary cardiac fibroblasts were isolated from endocardial biopsies from this cohort and shown to express increases in markers of fibroblast activation such as αSMA in response to TGF-β stimulation, although a comparison to control fibroblast activity was not performed [15]. Nonetheless, data from the studies presented here and those of others, suggest that increases in collagen accumulation in HFpEF appear to depend, at least in part, on decreases in collagen degradation.

The results presented herein suggest that cardiac fibroblasts from HFpEF hearts have distinct characteristics versus those isolated from failing hearts with a reduced ejection fraction (HFrEF). For example, fibroblasts isolated from patients with end-stage heart failure, primarily dilated cardiomyopathy (DCM) and ischemic cardiomyopathy (ICM), were recently reported to exhibit increases in proliferation versus fibroblasts from control hearts. Nagaraju *et al.* reported that primary fibroblasts from failing hearts could be separated into groups based on αSMA staining [16]. End-stage hearts contained increases in two proliferative myofibroblast (αSMA+) populations, identified by Ki67 positivity, over that of control hearts. In contrast, in our studies, fibroblasts from HFpEF hearts did not exhibit increases in proliferation and, in fact, trended toward decreased proliferation versus those of HTN (-) HFpEF [16]. In addition, Spruill et al. previously reported increases in MT1-MMP production in primary human cardiac fibroblasts isolated from patients with end-stage DCM presenting for heart transplant versus fibroblasts from referent control whereas we report herein a decrease in MT1-MMP by HTN (+) HFpEF fibroblasts [17]. ECM remodeling that takes place in DCM is predicted to be distinct from that of HFpEF. Hence, differences in fibroblast activity, *e.g.* proliferation and MT1-MMP activity, might be contributing factors to distinct myocardial structural changes indicative of HFpEF versus HFrEF.

## Limitations

One limitation to these studies is that primary cardiac preparations were established, cultured, and assessed for proliferation and protein production over a period of several years due to the availability of patient biopsies. Therefore, cultures were not able to be maintained in parallel. Hence, proliferation index were compared as percent changes from baseline so that each culture served as its own control. Similarly, protein production across multiple isolates were compared according to baseline control production of tubulin. The relatively small number of endpoints reported herein were limited by assays that were able to be carried out consistently over the time period of data collection. In addition, all primary cultures were plated on standard tissue culture plastic for consistency throughout this study. More recent data suggest that plating cardiac fibroblasts on substrates of significantly lower stiffness than tissue culture plastic are preferable for more accurately mimicking *in vivo* conditions. We predict that future studies in which primary human cardiac fibroblasts are plated on substrates that are more consistent with myocardial stiffness coefficients might yield greater differences in primary cell type phenotype between HTN (+) HFpEF versus referent control and HTN (-) HFpEF. Finally, although epicardial biopsies yield sufficient numbers of fibroblasts for *in vitro* study, it is appreciated that the due to the small size of the biopsies, primary fibroblast cultures might represent a subset of the fibroblast population in the heart. To this point, determinations of perivascular versus interstitial fibrotic deposition was not performed as sections from each biopsy did not necessarily contain equal vascularity in terms of blood vessel size and number.

## Supporting information

**S1 Fig. Collagen in cell layers: Western blot analysis of detergent soluble cell layers was carried out using anti-collagen I antibodies.** Values were normalized to levels of tubulin. A trend toward greater levels of collagen I in cell layers from HTN(+)HF fibroblasts was observed. Fibroblasts from P2 or P3 were used for analysis. Referent control (n = 4), HTN (-) HFpEF (n = 4), HTN(+) HFpEF (n = 6).
(PPTX)

## Author Contributions

**Conceptualization:** Yuhua Zhang, An O. Van Laer, Catalin F. Baicu, Lily S. Neff, Stanley Hoffman, Marc R. Katz, Sanford M. Zeigler, Michael R. Zile, Amy D. Bradshaw.

**Data curation:** Yuhua Zhang, An O. Van Laer, Catalin F. Baicu, Lily S. Neff, Stanley Hoffman, Marc R. Katz, Sanford M. Zeigler, Michael R. Zile, Amy D. Bradshaw.

**Formal analysis:** Yuhua Zhang, An O. Van Laer, Catalin F. Baicu, Lily S. Neff, Stanley Hoffman, Marc R. Katz, Sanford M. Zeigler, Michael R. Zile, Amy D. Bradshaw.

**Funding acquisition:** Lily S. Neff, Michael R. Zile, Amy D. Bradshaw.

**Investigation:** Catalin F. Baicu, Michael R. Zile, Amy D. Bradshaw.

**Methodology:** Yuhua Zhang, An O. Van Laer, Catalin F. Baicu, Lily S. Neff, Stanley Hoffman, Marc R. Katz, Sanford M. Zeigler, Michael R. Zile, Amy D. Bradshaw.

**Project administration:** An O. Van Laer, Catalin F. Baicu, Lily S. Neff, Michael R. Zile, Amy D. Bradshaw.

**Resources:** An O. Van Laer, Catalin F. Baicu, Michael R. Zile, Amy D. Bradshaw.

**Software:** Catalin F. Baicu, Michael R. Zile, Amy D. Bradshaw.

**Supervision:** An O. Van Laer, Catalin F. Baicu, Lily S. Neff, Michael R. Zile, Amy D. Bradshaw.

**Validation:** Yuhua Zhang, An O. Van Laer, Catalin F. Baicu, Lily S. Neff, Michael R. Zile, Amy D. Bradshaw.

**Visualization:** Catalin F. Baicu, Michael R. Zile, Amy D. Bradshaw.

**Writing – original draft:** Michael R. Zile, Amy D. Bradshaw.

**Writing – review & editing:** Yuhua Zhang, An O. Van Laer, Catalin F. Baicu, Lily S. Neff, Stanley Hoffman, Marc R. Katz, Sanford M. Zeigler, Michael R. Zile, Amy D. Bradshaw.

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
