## [Decision Letter · Decision Letter 0]

10 Nov 2021

PONE-D-21-32849Phenotypic Characterization of Primary Cardiac Fibroblasts from Patients with HFpEFPLOS ONE

Dear Dr. Bradshaw,

Thank you for submitting your manuscript to PLOS ONE. After careful consideration, we feel that it has merit but does not fully meet PLOS ONE’s publication criteria as it currently stands. Therefore, we invite you to submit a revised version of the manuscript that addresses the points raised during the review process.

We look forward to receiving your revised manuscript.

Kind regards,

Cristina Rodríguez Sinovas

Academic Editor

PLOS ONE

Journal Requirements:

a) Did participants provide their written or verbal informed consent to participate in this study?

5. Please ensure that you refer to Figure 3 in your text as, if accepted, production will need this reference to link the reader to the figure.

Reviewers' comments:

Reviewer's Responses to Questions

**Comments to the Author**

1. Is the manuscript technically sound, and do the data support the conclusions?

Reviewer #1: Yes

Reviewer #2: Yes

2. Has the statistical analysis been performed appropriately and rigorously? 

Reviewer #1: Yes

Reviewer #2: Yes

3. Have the authors made all data underlying the findings in their manuscript fully available?

Reviewer #1: Yes

Reviewer #2: Yes

4. Is the manuscript presented in an intelligible fashion and written in standard English?

Reviewer #1: Yes

Reviewer #2: Yes

5. Review Comments to the Author

Reviewer #1: The new clinical guidelines classify heart failure in three different groups (HFrEF; HFmrEF and HFpEF) based on the ejection fraction values. However, the underlying mechanism to understand the differences between the three groups remains unknown. Hence, this article point out some differences in fibroblast activity that I judge important for understanding heart failure. However, the following concerns must be considered.

1- As the authors describes, heart failure with a preserved ejection fraction (HFpEF) is characterized with an important increase in myocardial fibrosis. In the Figure 1, the authors show Picro Sirius representative images, the reference control and HTN (-) HFpEF evidence more interstitial fibrosis, in contrast the HTN (+) HFpEF image seems to present interstitial and perivascular fibrosis. The authors should specify whether there are differences in interstitial and perivascular fibrosis for the three groups showing images of interstitial/perivascular fibrosis. In addition, if there are differences or not you should discuss about their pathological meaning.

2- The authors calculate the collagen volume fraction according to Bradshaw, et al., 2009. As I have already commented above, due to the importance of the fibrotic process in HFpEF pathology, you should explain the collagen measurements in more detail (Figure 2). How many images you quantify for each group? Do you quantify different sections for each patient?

3- Please, explain in more detail the quantification of myocyte cross sectional area (how many fields of each sample).

4- The authors mention that they have observed no significant changes in cardiac fibroblast proliferation, but this may be can due to the low concentration of FCS (2%) used to stimulate proliferation. It is commonly used between 5-10% FCS to stimulate fibroblast proliferation (Lopez-Martinez, et al., 2019; Cañes L, et al., 2020). Moreover, the reference that the authors mention uses 10% FCS to stimulate fibroblast (Figure 4 of Bradshaw, et al., 1999). Do the authors analyze the impact of high FCS concentrations on proliferation assays?

5- Due to the partly insoluble character of collagen, western blotting analyses are not a reliable tool to determine the collagen production. The authors must confirm their western blotting results with other technics.

6- Please, improve the tubulin western blotting in figure 4 and 5.

Reviewer #2: In this study, the authors characterize the fibroblast phenotype in hypertensive patients with or without heart failure. The authors used epicardial biopsies from these patients to directly analyze fibroblast morphology and also performed fibroblast primary culture to study them in vitro. Although the data are interesting, revision of the experiments are needed in order to reach such conclusions.

Major concerns:

- My main concern is regarding the data obtained in the primary cell culture experiments. It has been shown (Cardiovasc Res. 2007 Aug 1;75(3):519-29; J Pathol. 2019 May;248(1):30-40.) that in primary fibroblast cell culture cells at passage 1 are regular fibroblast whereas cells at passage 3 are considered activated fibroblasts. Since here, some experiments were performed at passage 1 and others at passage 2 or 3, the results and experiments should be revised.

- Related to this, the passage of the cells should be clearly stated in figure legends and in the text.

6. PLOS authors have the option to publish the peer review history of their article (what does this mean?). If published, this will include your full peer review and any attached files.

Reviewer #1: **Yes: **Laia Cañes

Reviewer #2: No

---

## [Author Response · Author response to Decision Letter 0]

23 Dec 2021

We thank the editors and the reviewers for the helpful comments that improved our manuscript. We have addressed each reviewer concern individually and is included in the uploaded file, Response to Reviewers. As per journal instructions, we have 1) addressed PLOSOne style requirements, 2) clarified in our Methods that written consent was obtained from study participants, 3) included in the cover letter a repository URL containing blot images as raw data images, 4) included a full ethics statement in the ‘Methods’ section as requested, and 5) Fig. 3 is now cited in the text.

---

## [Editor Report · Decision Letter 1]

27 Dec 2021

Phenotypic Characterization of Primary Cardiac Fibroblasts from Patients with HFpEF

PONE-D-21-32849R1

Dear Dr. Bradshaw,

We’re pleased to inform you that your manuscript has been judged scientifically suitable for publication and will be formally accepted for publication once it meets all outstanding technical requirements.

Kind regards,

Cristina Rodríguez Sinovas

Academic Editor

PLOS ONE

Additional Editor Comments (optional):

The authors have satisfactorily answer all the queries of the reviewers
---

## [Editor Report · Acceptance letter]

3 Jan 2022

PONE-D-21-32849R1 

Phenotypic Characterization of Primary Cardiac Fibroblasts from Patients with HFpEF 

Dear Dr. Bradshaw:

I'm pleased to inform you that your manuscript has been deemed suitable for publication in PLOS ONE. Congratulations! Your manuscript is now with our production department. 

Kind regards, 

on behalf of

Dr. Cristina Rodríguez Sinovas 

Academic Editor

PLOS ONE